# ^1^H and ^31^P Magnetic Resonance Spectroscopic Metabolomic Imaging: Assessing Mitogen-Activated Protein Kinase Inhibition in Melanoma

**DOI:** 10.3390/cells13141220

**Published:** 2024-07-19

**Authors:** Pradeep Kumar Gupta, Stepan Orlovskiy, Fernando Arias-Mendoza, David S. Nelson, Kavindra Nath

**Affiliations:** 1Molecular Imaging Laboratory, Department of Radiology, University of Pennsylvania, Philadelphia, PA 19104, USA; stepan.orlovskiy@pennmedicine.upenn.edu (S.O.); fernando.arias-mendoza@pennmedicine.upenn.edu (F.A.-M.); dsnelson@pennmedicine.upenn.edu (D.S.N.); 2Advanced Imaging Research, Inc., Cleveland, OH 44114, USA

**Keywords:** melanoma, trametinib, ^1^H/^31^P magnetic resonance spectroscopy, oxygen consumption rate, extracellular acidification rate, glucose uptake, lactate production

## Abstract

The MAPK signaling pathway with *BRAF* mutations has been shown to drive the pathogenesis of 40–60% of melanomas. Inhibitors of this pathway’s BRAF and MEK components are currently used to treat these malignancies. However, responses to these treatments are not always successful. Therefore, identifying noninvasive biomarkers to predict treatment responses is essential for personalized medicine in melanoma. Using noninvasive ^1^H magnetic resonance spectroscopy (^1^H MRS), we previously showed that BRAF inhibition reduces lactate and alanine tumor levels in the early stages of effective therapy and could be considered as metabolic imaging biomarkers for drug response. The present work demonstrates that these metabolic changes observed by ^1^H MRS and those assessed by ^31^P MRS are also found in preclinical human melanoma models treated with MEK inhibitors. Apart from ^1^H and ^31^P MRS, additional supporting in vitro biochemical analyses are described. Our results indicate significant early metabolic correlations with response levels to MEK inhibition in the melanoma models and are consistent with our previous study of BRAF inhibition. Given these results, our study supports the potential clinical utility of noninvasive MRS to objectively image metabolic biomarkers for the early prediction of melanoma’s response to MEK inhibition.

## 1. Introduction

Melanoma, the deadliest form of human skin cancer, originates from melanocytes and exhibits a high tendency to metastasize, accounting for approximately 90% of skin cancer-related mortality [1]. Melanoma represents 1.7% of all global cancer diagnoses and ranks as the fifth most common cancer in the United States [2]. While melanoma is highly treatable when confined to its primary site, metastatic melanoma presents a grim outlook, with a median survival of only approximately six months [3,4,5]. Furthermore, systemic therapies currently employed in patients with metastatic melanoma exhibit varying response rates, with the rapid development of tumor resistance observed in the majority of patients [3,6,7]. While surgical intervention remains the primary treatment for melanoma, recent breakthroughs in immunotherapy and targeted molecular therapies for metastatic melanoma offer significant potential [8].

Melanoma treatment, incorporating surgery, chemotherapy, immunotherapy, and radiotherapy, faces challenges due to melanoma resistance, primarily attributed to melanin production. Recently, the focus has been on targeting the mitogen-activated protein kinase (MAPK) pathway to overcome resistance and improve therapeutic outcomes [9,10]. Two signaling targets along the MAPK pathway in treating melanoma are mutated serine/threonine-protein kinase B-Raf (BRAF) [11,12] and mitogen-activated protein kinase kinase (MEK) [13]. Inhibiting the BRAF or MEK portions of the MAPK pathway exerts primarily cytostatic effects compared to chemotherapy.

MEK inhibition in BRAF mutant cells may induce apoptotic and cytostatic impacts [14]. Early noninvasive biomarkers are crucial to assessing target modulation and treatment efficacy due to the potential absence of tumor shrinkage. BRAF is highly expressed in melanocytes, neural tissue, testes, and hematopoietic cells [15]. BRAF is activated by RAS-specific GTPases (i.e., rat sarcoma virus-specific guanosine triphosphate hydrolases) expressed in all animal cell lineages and organs [16]. Phosphorylated BRAF activates MEK (a kinase component of the MAPK pathway), which, in turn, activates extracellular signal-regulated kinases (ERK-MAPK) by phosphorylation and thus stimulates growth and transformation [15]. Unfortunately, melanoma cells have a hypermutable genome leading to tumor resistance by responding to the blockage of the MAPK pathway by rerouting to an alternate path. Therefore, most patients develop resistance to targeted therapy within weeks to months of the initiation of treatment.

Melanoma often exhibits a glycolytic phenotype driven by the constitutive activation of the *BRAF* gene mutation (Figure 1) [17]. This activation increases glucose uptake and glycolysis, mediated by the MAPK pathway and its induction of hypoxia-inducible factor 1α (HIF-1α), a key regulator of glycolytic activity [18,19]. In response to varying energy demands and environmental cues, melanomas demonstrate metabolic plasticity, adjusting between glycolysis and oxidative phosphorylation (OXPHOS) [20]. Glutamine could also become a primary energy source, facilitated by heightened glutaminolysis and the upregulation of glutamine transporters. Furthermore, resistance to BRAF and MEK inhibition primarily arises from metabolic adaptations within the MAPK signaling pathway [20,21].

Published research, including ours, shows that magnetic resonance spectroscopy (MRS) is a noninvasive avenue for monitoring metabolic changes in melanoma models following diverse anticancer treatments [10,14]. The present study focuses on identifying metabolic biomarkers of a response to trametinib in melanoma models using MRS. Similar to MR imaging (MRI), spatial localization pulses can be added while acquiring MRS. Techniques exist to acquire MRS from a single volume or several volumes in one, two, or three dimensions, making MRS an appropriate imaging technique (i.e., MR spectroscopic imaging or MRSI). However, using single-volume methods, as in the present report, allows us to pinpoint the specific localization of the metabolic information but prevents us from producing a visual representation of the data (i.e., image). We utilized single-voxel MRS to detect and quantify tumor metabolites that indicate a treatment response specifically located in the tumor mass.

We demonstrated that MRS discerns significant changes in metabolic signatures in successfully treated melanoma models with dabrafenib, a BRAF inhibitor, showcasing early therapy-related changes in metabolomics, pH, and bioenergetics [10]. This elucidation underscores the potential of MRS in delineating early-treatment metabolic responses to targeted therapies. Notably, the MRS availability in clinical MRI systems guarantees the translation of the metabolic signatures as possible biomarkers of treatment efficacy, supporting personalized therapeutic approaches in clinical melanoma management.

This report examines four human melanoma cell lines characterized by their BRAF sensitivity: a wild-type (WM3918), a BRAF-resistant mutant-type (WM983BR), and two BRAF-sensitive mutant-types (WM983B and DB-1). These studies tested the cell lines’ response to trametinib, a MEK inhibitor targeting the MAPK signaling pathway (Figure 1). The inhibition of MEK is known to elicit treatment responses in melanoma models, as the hyperactive BRAF precedes MEK in the MAPK pathway. To discern trametinib-induced metabolic changes, we grew the melanoma cell lines in athymic nude mouse xenografts and cell cultures. We employed in vivo ^1^H and ^31^P MRS noninvasively in the xenografts alongside other in vitro analytical methods in the cell cultures to assess the metabolic response of human melanoma.

**Figure 1 cells-13-01220-f001:**
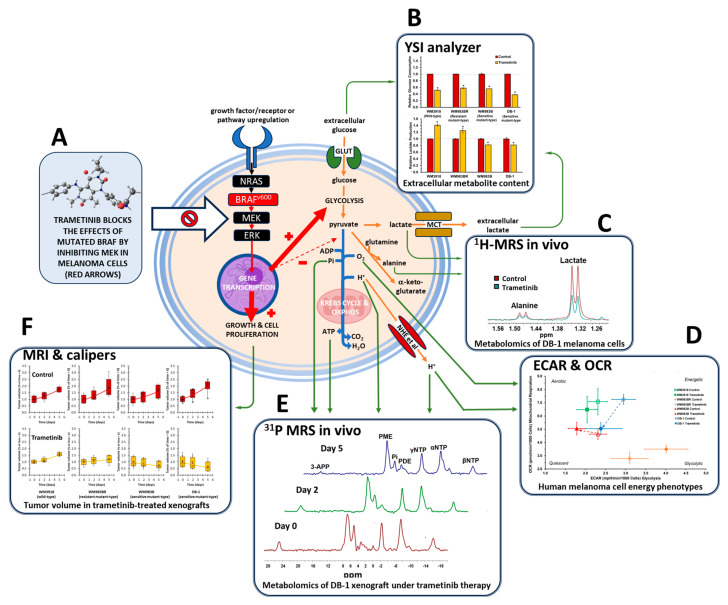
Schematic representation of the metabolomics research in human melanoma models treated with trametinib. The red arrows highlight the effects of the mutated BRAF protein (i.e., increased growth and cell proliferation, increased glycolysis, and reduced pyruvate oxidation). Trametinib (structure in (**A**); from istrockphoto.com, accessed on 18 June 2024) exerts its effect, inhibiting MEK and, thus, blocking the mutated BRAF effects (white block arrow). Mutated BRAF enhances the metabolic pathways highlighted with orange arrows, while those inhibited are shown with blue arrows. The green arrows indicate the procedures for determining the highlighted metabolites or cellular processes (Figure 2, Figure 3, Figure 4, Figure 5, Figure 6 and Figure 7). In summary, we measured extracellular glucose and lactate levels (**B**), intracellular lactate and alanine levels via noninvasive ^1^H MRS (**C**), the effect of trametinib on the extracellular acidification rate (ECAR), and oxygen consumption rate (OCR; (**D**)), bioenergetic parameters and intra- and extracellular pH via noninvasive ^31^P MRS (**E**), and tumor growth (**F**). An asterisk (*) indicates a statistically significant difference (*p* < 0.05) between control (*n* = 3) and trametinib-treated (*n* = 3) groups.

## 2. Materials and Methods

### 2.1. Reagents and MEK Inhibitor

The reagents used for cell cultures and biochemical analyses, including 3-aminopropylphosphonate (3-APP) and their sources, are listed in our previous study [10]. Trametinib was obtained from LC Laboratories (Woburn, MA, USA). Trametinib was dissolved in 0.5% hydroxypropyl methylcellulose and 0.2% Tween 80 buffer solution in Milli-Q filtered water as the vehicle for mouse injection. Dimethylsulfoxide (DMSO; Millipore Sigma, St. Louis, MO, USA), when tested at the levels used as a vehicle for in vitro studies, did not affect the cell growth or metabolic signatures of the melanoma models studied.

### 2.2. Cell Lines and Cell Culture Conditions

Briefly, WM3918 (Accession No: CVCL_C279) and WM983BR (Accession No: CVCL_AP81) melanoma cells were cultured in MCDB153/L-15 medium supplemented with antibiotics, calcium chloride, and 2% FBS. WM983BR medium also contained 100 nM dabrafenib. DB-1 (Accession No: CVCL_D902) and WM983B (Accession No: CVCL_6809) cells were cultured in MEM-α medium supplemented with glucose, glutamine, antibiotics, MEM NEAA, and FBS. More details of the culturing conditions are described elsewhere [10].

### 2.3. Extracellular Glucose and Lactate Measurements

Melanoma cell lines were seeded in T-75 tissue culture flasks and incubated at 37 °C with 5% CO_2_ for 24 h to allow adhesion. Afterward, either vehicle (0.1% DMSO) or 7 nmol/L trametinib was added to create control and treated groups. After 48 h of treatment, medium samples were collected, and extracellular glucose and lactate concentrations were measured using a YSI 2300 STAT PLUS Biochemistry Analyzer (YSI Incorporate-Yellow Springs-Xylem, Yellow Springs, OH, USA).

### 2.4. Mitochondrial Stress and Cell Energy Phenotype Assays

Immediately before the assays, the cell lines were incubated in the presence of either 0.1% DMSO (control) or 7 nmol/L trametinib in 0.1% DMSO for 48 h. Cell mitochondrial stress and energy phenotype assays were carried out using a Seahorse XFe96 Extracellular Flux Analyzer (Agilent Technologies; Santa Clara, CA, USA). Melanoma cells were seeded in Seahorse 96-well cell culture microplates at 15–20 × 10^3^ cells per well and incubated for 24 h before the assay was started. Fresh stock solutions of oligomycin (100 µM), FCCP (100 µM), and rotenone/antimycin-A (50 µM) were prepared on the assay day. Using these stocks, we prepared working oligomycin, FCCP, and rotenone/antimycin-A solutions at 15 µM, 5 µM, and 5 µM, respectively. The final concentration of these reagents in the microplate wells was 1/10 of the working solutions. Additionally, 30 µL of 20 mM Hoechst 33,342 solution (Thermo Scientific, Waltham, MA, USA) was added to the rotenone/antimycin-A solution to stain the cells for cell counting. These experiments followed the user guide of the flux analyzer. Oxygen consumption rates (OCR) and extracellular acidification rates (ECAR) were determined for each of the four melanoma lines and used to create a cell energy phenotype map.

After the assays, the microplates were transferred to a BioTek Cytation 5 cell imaging multimode reader (Agilent Technologies) using fluorescence scanning for cell count and viability. The OCR and ECAR data were normalized to viable cell count for comparisons.

### 2.5. In Vitro Measurement of Intracellular Metabolites by High-Resolution ^1^H-MRS

We conducted ^1^H-MRS on extracted melanoma cells grown in T-182 tissue culture flasks. The cells were grown in either vehicle (0.1% DMSO) or 7 nmol/L trametinib in 0.1% DMSO for 48 h. After treatment, 10–15 × 10^6^ cells were harvested, centrifuged, and washed with cold phosphate-buffered saline (PBS), then stored at −80 °C. Frozen cell pellets were resuspended in an 80% methanol--water solution, homogenized, sonicated, and centrifuged to extract metabolites. The supernatant containing the metabolites was lyophilized and resuspended in deuterium oxide (D_2_O) with trimethylsilyl propanoic acid (TSP) and transferred to a 5 mm NMR tube. This method is described in detail elsewhere [10]. High-resolution ^1^H-MRS spectra were acquired using a PRESAT pulse sequence (water suppression with presaturation pulses) on a 9.4 T/8.9 cm vertical bore Varian NMR spectrometer, with a flip angle of 45°, repetition time (TR) of 8.8 s, spectral width (SW) of 6756.8 Hz, 16,384 points, and 128 transients. MestRec 6.1 software (Mestrelab Research, Santiago de Compostela, Spain) was used for data processing. Spectra were filtered with a 1 Hz exponential function to improve the apparent signal-to-noise ratio (SNR). Peak areas were integrated and normalized by the TSP signal and the number of protons in the signal to obtain molar concentrations.

### 2.6. Mouse Preparation for Proton (^1^H) and Phosphorus (^31^P) MRS Studies

The MRS study cohort comprised 48 human melanoma-bearing male athymic nude mice obtained from Charles River Laboratories (https://www.criver.com/, accessed on 18 June 2024). The mice were divided into four groups, each xenografted with a different human melanoma cell line. Before injection, cells were cultured in a monolayer at 37 °C in a controlled environment with 5% CO_2_ using Hank’s Balanced Salt Solution (HBSS). This controlled culture setup allowed the cells to grow and multiply until they reached the desired density for injection. Animals were injected subcutaneously into their right thigh with 0.1 mL of a 10^9^/mL suspension of the melanoma cells in HBSS. After injection, the xenografted tumors were allowed to grow until their volume reached approximately 250 mm^3^. The tumors were permitted to establish themselves in the host mouse to develop characteristics like those in human melanoma during their initial growth period, providing a relevant preclinical model for studying the disease and potential treatments.

Subsequently, each group of mice was divided into two subgroups. One received the vehicle (placebo), and the other trametinib (10 mg/kg), once daily through oral gavage for five days. The number of mice in groups WM3918 and WM983BR was *n* = 5 for the placebo and trametinib subgroups, while in groups WM983B and DB-1 the number was *n* = 6 for the placebo and *n* = 8 for the trametinib subgroups.

### 2.7. Experiments Involving In Vivo ^1^H and ^31^P MRS

In vivo MRS exams were conducted noninvasively in each mouse on Day 0, Day 2, and Day 5 regarding the start of placebo or trametinib administration. For these exams, mice were anesthetized with 1% isoflurane in O_2_/CO_2_ (95%/5%) at a rate of 1 L/min. To measure extracellular pH (pHe) during ^31^P MRS, 3-APP (300 mg/mL solution in water) was injected into the mouse peritoneum before placing it in the spectrometer. The animal’s core temperature (37 ± 1 °C) and breathing were monitored during the experiment.

In vivo, MRS exams were performed on a 9.4 T Bruker horizontal bore spectrometer (Bruker Instruments, Billerica, MA, USA). ^1^H MRS was carried out using a slice-selective, double-frequency, Hadamard-selective, multiple quantum coherence (HDMD-Sel-MQC) transfer pulse sequence to acquire the methyl ^1^H signals of lactate and alanine selectively [22]. For these studies, the tumor was positioned in a homemade single-frequency (^1^H) slotted-tube resonator (15 mm outer diameter, 13 mm inner diameter, 16.5 mm depth). The following acquisition parameters were used: TR = 4 s, 1000 points, and 32 transients. A localized water spectrum was also acquired using the same sequence without water suppression (TR = 4 s, 4 transients).

^31^P MRS exams were performed as described in detail previously [23]. In summary, the tumor was placed in a homemade dual-frequency (^1^H/^31^P) slotted-tube resonator with a diameter of 10 mm. The Image-Selected in vivo Spectroscopy (ISIS) pulse sequence was employed with the following parameters: acquisition time (AT) of 64 ms, flip angle = 90°, TR = 2 s, SW = 7979 Hz, offset frequency = 430, and 512 points. ^31^P MRS spectra were obtained using a decoupling duration of 64.16 ms, a decoupling power of 3.12 W, and a CPD sequence element of 0.5 ms. A scout image was initially acquired to set the acquisition voxel to minimize the contamination of tumor metabolite signals with exogenous signals from lipids and muscle.

All spectroscopic data were processed using the NUTS (Acorn NMR Inc., Livermore, CA, USA) and MestRec programs. We applied 10 and 40 Hz exponential filters to enhance the apparent SNR of ^1^H and ^31^P MRS, respectively. Baseline correction was performed before plotting and calculating peak areas. In the ^1^H spectra acquired, the ^1^H methyl signals from lactate and alanine were integrated. In addition, the unsuppressed water spectrum was used to integrate the ^1^H signal from water (H_2_O) to obtain the in vivo lactate/H_2_O and alanine/H_2_O values in the tumors. Similarly, the ^31^P signals from the center phosphate of NTP (i.e., β-NTP) and inorganic phosphate (Pi) were integrated to obtain the β-NTP/Pi ratio. Furthermore, the centroid of the Pi, 3-APP, and α-NTP signals was measured, and the pH values of the intracellular (chemical shift difference between Pi and α-NTP) and extracellular compartments (chemical shift difference between 3-APP and α-NTP) were calculated using adaptations of the Henderson–Hasselbalch equation [24].

### 2.8. Measurement of Tumor Volume

Tumor dimensions in the mouse xenografts were assessed using calipers. Measurements were taken in three orthogonal directions, and tumor volume was calculated using the formula V = π (l × w × d)/6, where l = length, w = width, and d = depth of the tumor.

### 2.9. Statistical Analysis

The data are presented graphically and summarized using either bar graphs denoting the mean ± standard deviation (σ ± SD—Figure 2, Figure 3 and Figure 4 and Table 1) or box-and-whiskers plots showing the median, mean, quartiles, data points, and outliers (Figure 5, Figure 6 and Figure 7). Two-tailed Student’s *t*-tests assuming variance homogeneity were conducted to compare untreated (control) vs. trametinib-treated pairs, setting the significance level at α = 0.05. In Figure 5, Figure 6 and Figure 7, the data are presented as time-related changes. Although exponential behavior is expected for biological data like those presented (e.g., tumor growth), we opted to use linear regression due to the limited timespan of the studies and the fact that the linear and exponential regressions reported *R*^2^ values within 95% of each other. We assumed linearity and variance homogeneity for these graphs, and outliers were included in the analyses presented. The adjusted *R*^2^ and *p*-values of each time course were determined (Figure 5, Figure 6 and Figure 7), but we only show the adjusted *R*^2^ values with a substantial effect size (adjusted *R*^2^ ≥ 0.26). Weak or moderate effect sizes (adjusted *R*^2^ < 0.26) are marked in the tables with the word “low”, and non-significant *p*-values (>α) with the characters “N.S.” (non-significant) in red. Tumor volume was normalized for individual tumors by the mean volumes for Day 0 in all groups to obtain the relative tumor volume (Figure 7). These mean values were then averaged across tumors to obtain mean values for individual days.

## 3. Results

### 3.1. Trametinib’s Impact on Glucose Consumption and Lactate Production in Melanoma In Vitro

Extracellular glucose and lactate concentrations were assessed using a YSI 2300 STAT PLUS Glucose Lactate Analyzer, unveiling significant variations in response between untreated cells and those subjected to trametinib treatment (7 nM for 48 h), as depicted in Figure 2. Specifically, a significant decrease in glucose consumption was observed in all cell lines in the presence of trametinib. Furthermore, lactate production was increased in the wild-type WM3918 and BRAF-resistant WM983BR melanoma cell lines. In contrast, the sensitive human melanoma cell lines (WM983B and DB-1) exhibited a decrease in extracellular lactate production. This divergence in lactate production highlights the distinct metabolic phenotype of these cell lines affecting their response to trametinib treatment. 

**Figure 2 cells-13-01220-f002:**
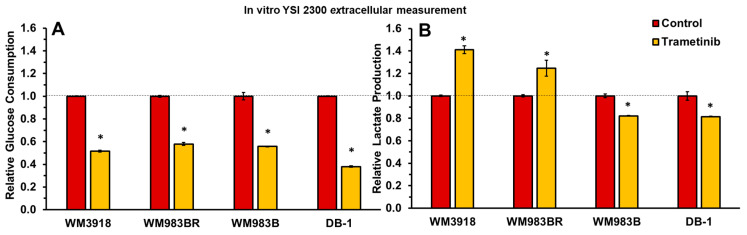
Extracellular glucose consumption (**A**) and lactate production (**B**) were measured in the four cultured melanoma cell lines in the absence (control) or presence of trametinib (7 nmol/L) for 48 h. Relative values (y-axes) were obtained by dividing the extracellular metabolite content by the number of cells. Error bars denote standard deviation (SD). An asterisk (*) indicates a statistically significant difference (*p* < 0.05) between control (*n* = 3) and trametinib-treated (*n* = 3) groups. The dashed line indicates the relative values of the controls.

### 3.2. Oxygen Consumption and Extracellular Acidification Rates In Vitro

As shown in Figure 3 and Table 1, WM3918 cells treated with trametinib showed no significant differences in OCR or the OCR/ECAR ratio but a statistically significant decrease in ECAR relative to their control values. Conversely, trametinib-treated WM983BR cells showed significant shifts in OCR and ECAR vs. control cells. Still, the resulting shifts in this cell line’s OCR/ECAR ratio were insignificant, probably due to the high dispersion of the measurements.

In comparison, trametinib-treated WM983B and DB-1 cells showed substantial shifts in the OCR/ECAR ratio compared to control cells. Trametinib-treated WM983B cells showed decreased ECAR and increased OCR, while treated DB-1 cells showed an overall decrease in energy (i.e., reduced OCR and ECAR). Significant shifts occurred in the OCR/ECAR ratio of the DB-1 and WM983B cells treated with trametinib (Figure 3 and Table 1). However, while DB-1 cells were significantly de-energized by trametinib (i.e., a drop in OCR and ECAR, blue arrow), WM983B cells showed a more respiratory phenotype after trametinib (i.e., decreased ECAR and increased OCR, red arrow).

**Table 1 cells-13-01220-t001:** Effect of trametinib in OCR, ECAR, and the OCR/ECAR value in melanoma cell lines. Data are shown as mean value ± standard deviation with *n* = 8 per group. We used the Student independent *t*-test to determine statistical significance.

Group	Mean ± SD (*n* = 8)		*p*-Value
	Control	Trametinib	
Oxygen consumption rate (pmol/min/1000 cells)			
WM3918	7.09 ± 0.99	6.49 ± 1.01	0.13
WM983BR	2.79 ± 0.50	3.50 ± 0.31	0.007
WM983B	4.63 ± 0.41	5.06 ± 0.48	0.014
DB-1	7.25 ± 0.36	5.05 ± 0.46	<0.001
Extracellular acidification rate (pmol/min/1000 cells)			
WM3918	2.31 ± 0.24	2.04 ± 0.26	0.011
WM983BR	3.10 ± 0.46	4.01 ± 0.54	0.015
WM983B	2.31 ± 0.22	1.78 ± 0.21	0.001
DB-1	2.95 ± 0.29	2.38 ± 0.53	0.001
OCR/ECAR ratio—cell energy phenotype			
WM3918	3.08 ± 0.37	3.18 ± 0.29	0.46
WM983BR	0.90 ± 0.12	0.88 ± 0.08	0.71
WM983B	2.02 ± 0.22	2.89 ± 0.55	<0.001
DB-1	2.48 ± 0.28	2.17 ± 0.33	0.01

**Figure 3 cells-13-01220-f003:**
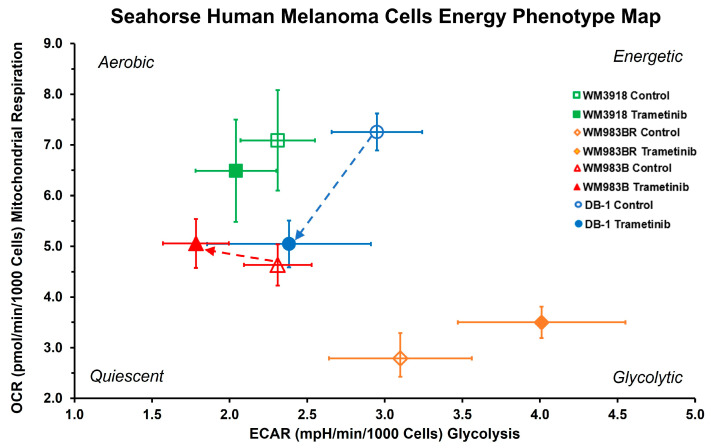
Effect of trametinib treatment on OCR and ECAR of cultured melanoma cells. The graph shows the four melanoma cell lines’ OCR vs. ECAR under basal conditions. Open symbols represent untreated cells, while solid symbols represent trametinib-treated cells. The dashed arrows designate cell lines with statistically significant shifts in their OCR/ECAR ratio (Table 1), which indicates different cell energy phenotypes and shows shifts in the preference of energy pathways for each cell line under trametinib.

### 3.3. In Vitro ^1^H MRS to Test the Metabolic Response to Trametinib in Isolated Melanoma Cells

We conducted ^1^H MRS at 9.4T in suspensions of the four melanoma cell lines to determine the metabolic effect of trametinib. For these experiments, untreated and trametinib-treated cells (7 nM for 48 h) were studied. As shown in Figure 4, we focused on lactate and alanine, as these metabolites relate to the Warburg effect, abnormal metabolic reprogramming in cancer that affects glycolysis [25,26,27,28].

**Figure 4 cells-13-01220-f004:**
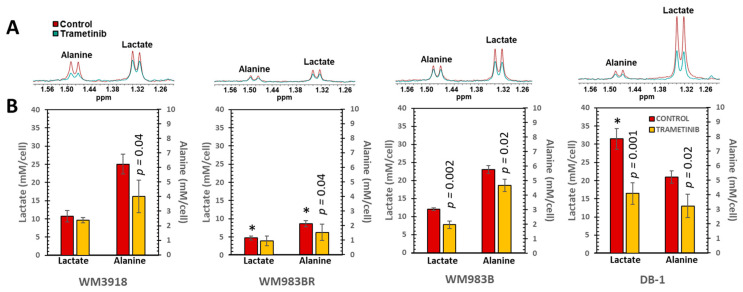
In vitro ^1^H MRS of human melanoma cell lines. (**A**) Representative high-resolution ^1^H MRS spectra depicting alanine and lactate in the four cell lines are illustrated. Control spectra are displayed in red, while trametinib-treated spectra are shown in green. (**B**) Mean values ± SD (*n* = 3) of the intracellular lactate and alanine concentrations were determined by integrating their ^1^H MRS signals in (**A**) and normalized by the trimethylsilyl propanoic acid (TSP) content protons to obtain mM/cell concentrations. Asterisks (*) denote a statistically significant difference (*p* < 0.05) between the control values of lactate or alanine in the WM3918 cell line (in red) and the control values in the other three cell lines. The *p*-values on top of the trametinib-treated data (in yellow) demonstrate the statistical significance in the trametinib-related reduction vs. its control value.

Figure 4 shows that the lactate levels in the untreated (control) cultured melanoma cells varied substantially. Student t-tests indicated that the lactate values of untreated cells were significantly different when the WM3918 cell line was compared with WM983B and DB-1 but not with WM983BR (asterisks on top of red bars in Figure 4). Conversely, when comparing WM3918 with the rest of the cell lines, only the WM983BR cell line showed a significantly reduced alanine value. Furthermore, trametinib-treated WM3918 and WM983BR cells did not show statistically significant changes in lactate vs. their untreated counterparts. However, this comparison was significant for the lactate reductions in the WM983B and DB-1 cell lines. Moreover, trametinib-related changes in alanine were significant for all cell lines except WM983BR (*p*-values on top of yellow bars in Figure 4.

### 3.4. In Vivo ^1^H- and ^31^P- MRS of Melanoma Xenografts in Mice

The trametinib treatment’s effect on the metabolomics of human melanoma xenografts in mice was noninvasively tested using MRS. The melanoma implants on the mice’s flanks underwent in vivo ^1^H and ^31^P MRS evaluation before (Day 0) and during trametinib therapy (Day 2 and Day 5).

Figure 5 shows trametinib’s impact on intracellular lactate and alanine levels tracked noninvasively in melanoma xenografts. The HDMD-Sel-MQC transfer pulse sequence facilitated the selective detection and integration of the lactate and alanine methyl resonances. The water signal from the acquired unsuppressed spectrum was also integrated and used to normalize the lactate and alanine signals. The lactate/H_2_O [29] and alanine/H_2_O ratios were considered representative of the intracellular metabolites’ molar concentrations. Figure 5 shows that untreated mice have a significant, time-related lactate and alanine ratio increase regardless of cell line (panels B and C, control row in red), with a substantial effect size (i.e., *R*^2^ ≥ 0.26) and significant regression in all cell lines. In comparison, the tumor lactate and alanine values in treated mice (panels A and B, trametinib row in yellow) show striking differences. While the WM3918 melanoma cell line exhibited similar lactate and alanine time-related increases to those in its control group (with a substantial effect size and *p*-value), the rest of the cell lines did not show metabolite increases during therapy. Instead, the WM983BR cell line showed no time dependency for the lactate and alanine tumor content (i.e., near-zero slope, low adjusted *R*^2^, and no significance on the linear regression analysis), while WM983B and DB-1 showed significant time-related decreases in the metabolites’ tumor content. Furthermore, the WM3918 cell line did not show statistical differences in lactate or alanine between the control and trametinib-treated groups upon comparing the slopes or group means of each day. Regarding lactate, the results in the WM3918 xenografts match the results in the isolated cells. Still, we did not reproduce the significant trametinib-related reduction in alanine in isolated cells of this cell line (Figure 4). In addition, the variability in lactate and alanine found in untreated cultured cells was not reproduced in any of the three studies of the untreated (control) xenografts, except for an increase in lactate in the untreated DB-1 xenografts.

**Figure 5 cells-13-01220-f005:**
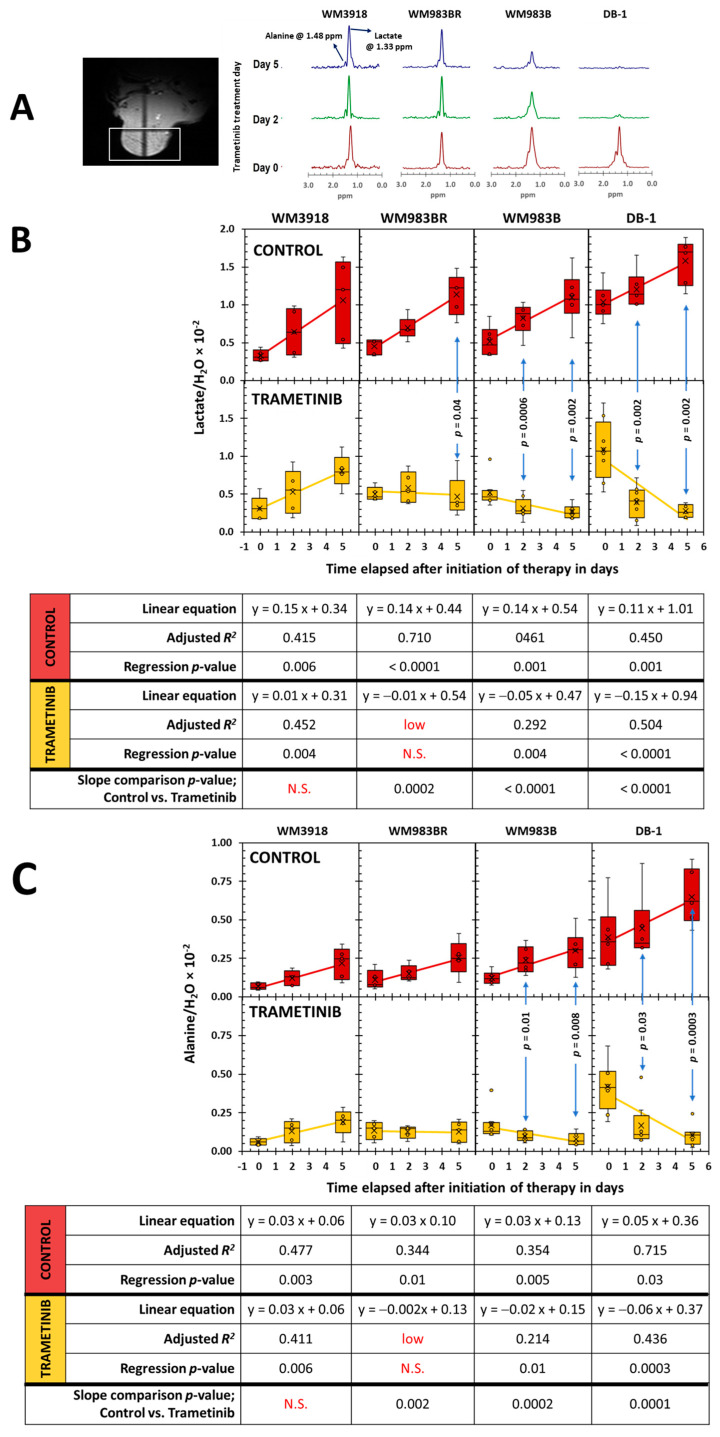
In vivo localized ^1^H MRS results. (**A**) Localizer tumor images for noninvasive ^1^H MR spectra were acquired using the Hadamard-selective MQC transfer pulse sequence to measure lactate and alanine. The spectra at Day 0 (red), Day 2 (green), and Day 5 (blue) of trametinib therapy in each xenograft model are shown from bottom to top. Box-and-whiskers plots of time-related in vivo changes are shown in (**B**) for tumor lactate and (**C**) for alanine in mouse xenografts of each cell line. At the top of (**B**,**C**), in red, the box plots show the lactate and alanine changes in the untreated (control) groups, while at the bottom, in yellow, the lactate and alanine changes in the trametinib-treated mice are shown. The *p*-values in (**B**,**C**) denote the statistical significance of the mean difference in lactate (**B**) and alanine (**C**) for the control vs. trametinib groups at each time point. The tables below (**B**,**C**) summarize the time-related analysis of each metabolite using linear regressions. The equations that describe the time course, the adjusted *R*^2^, and the significance of the regression are shown for the control (red) and trametinib time curves (yellow). The last row in both tables shows the statistical significance when the slopes of the control vs. trametinib are determined. Notation in red denotes the adjusted *R*^2^ < 0.2 (low) or *p*-values that are non-significant (N.S.).

Furthermore, the time-related differences in the tumor lactate and alanine content between the untreated and trametinib-treated groups in the WM983BR, WM983B, and DB-1 xenografts were highly significant (slope comparison *p*-value in Figure 5). Figure 5 shows statistical differences between the control vs. trametinib pairs in these cell lines on Day 2 and Day 5, except for Day 2 of WM983BR. Given that this late appearance of a significant lactate reduction in the trametinib-treated WM983BR xenografts (until Day 5) could be the reason for the non-significant lactate reduction found in the isolated cells of treated WM983BR, we can conclude that the trametinib effect on qualitative metabolic changes (or lack thereof) in the cultured cells and xenografts of the four melanoma cell lines match.

Figure 6A displays the in vivo ^31^P MRS of the melanoma models, demonstrating distinct low-field resonances, including the NTP’s three signals, Pi, and the exogenously added 3-APP. Given that the most concentrated NTP in biological tissues is adenosine triphosphate (ATP) and that there is a rapid exchange of the terminal phosphates of all the NTPs, we considered the NTP/Pi ratio to be an acceptable measure of the cellular ATP hydrolytic state, a reliable indicator of bioenergetic status. We calculated the NTP/Pi ratio by integrating the β-signal of NTP because it does not overlap with other signals compared to the α- and γ-NTP signals.

Figure 6B shows that time changes in β-NTP/Pi were not demonstrated in control mice regardless of the cell line (near-zero slope, low adjusted *R*^2^, and *p* > 0.05). In comparison, except for the trametinib-treated WM3918, the remaining three xenografts showed significant time-related increases in the β-NTP/Pi ratio with substantial effect sizes (adjusted *R*^2^ ≥ 0.026). Again, excluding the WM3918 data, comparing the control vs. trametinib-treated time courses of β-NTP/Pi in the remaining xenografts is significant. In addition, comparing the control vs. trametinib pairs at each time point shows significance for Day 2 and Day 5, except for Day 2 of WM983BR.

Figure 6C shows the pHi values and Figure 6D the pHe values determined by ^31^P MRS. Only DB-1 xenografts exhibited a significant time-dependent decrease in pHi under trametinib treatment. In addition, the untreated xenografts of the same DB-1 xenografts showed significant pHe reductions but no time change during treatment. Therefore, the time course comparison of the pHe data in the control vs. trametinib groups of the DB-1 xenografts is statistically significant.

**Figure 6 cells-13-01220-f006:**
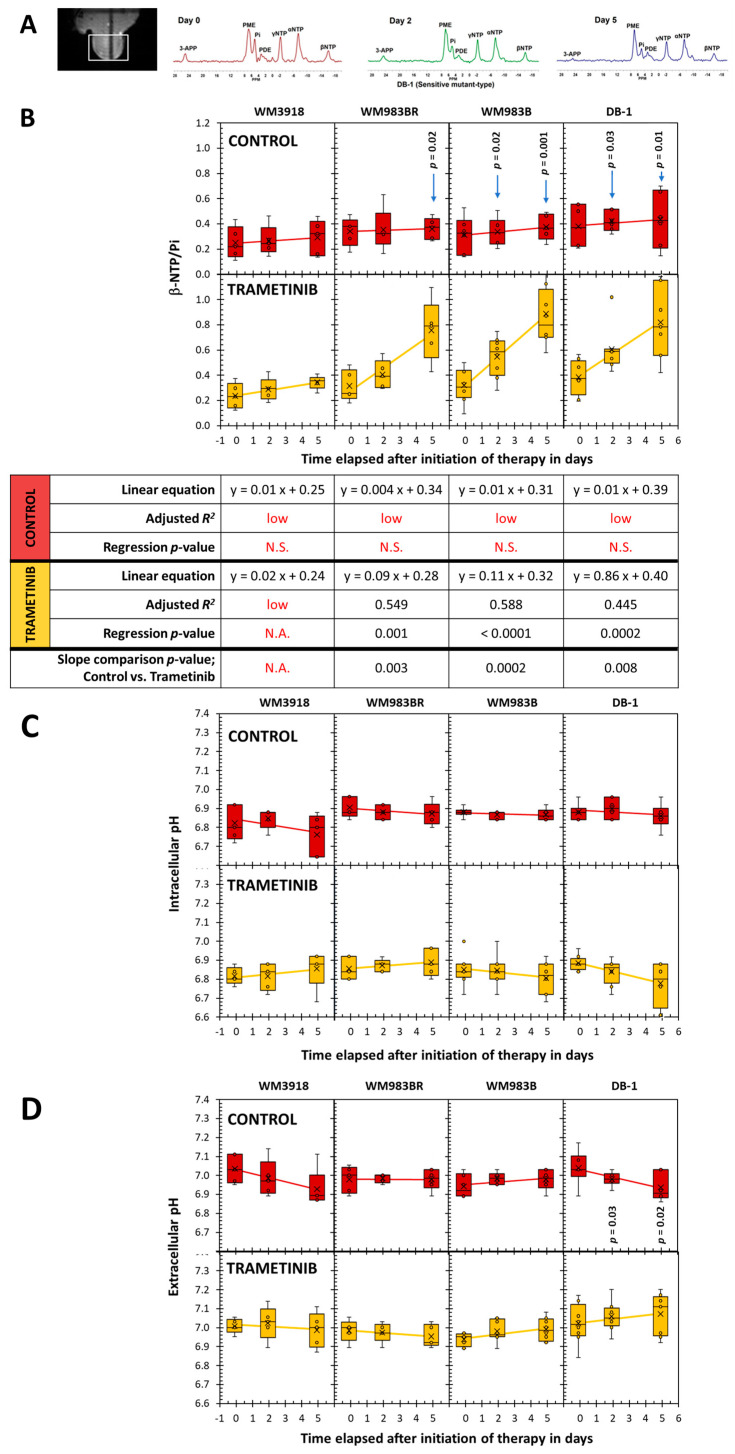
In vivo localized ^31^P MRS results. (**A**) Localizer tumor images for noninvasive MR spectra were acquired using the ISIS pulse sequence of DB-1 human melanoma xenografts at Day 0 (red), Day 2 (green), and Day 5 (blue) of trametinib treatment. Peak assignments: 3-APP, 3-amino propylphosphonic acid; PME, phosphomonoesters; Pi, inorganic phosphate; PDE, phosphodiesters; NTP, the α, β, and γ ^31^P signals of nucleoside triphosphates. We determined the chemical shift differences between Pi and α-NTP and those of 3-APP and α-NTP to obtain the pH values of the intracellular (pHi) and extracellular compartments (pHe), respectively. The intensity variability of 3-APP shown is due to factors like animal weight and administration method, but does not interfere with the pHe measurements. (**B**) Box-and-whiskers plots of the time-related changes in the β-NTP/Pi ratio measured in the ^31^P MR spectra. In red, the top row of box plots depicts the data of untreated mice set apart by the xenografted cell line, while on the bottom row of box plots, in yellow, the trametinib-treated results are shown. The *p*-value of the mean comparison of the control vs. treated pair at each time point is shown in the graphs. Like in Figure 5, the bottom table in (**B**) summarizes the fitting analysis of the data to time-related linear regressions and the slope comparisons between control vs. trametinib-treated cells in each cell line. The pH values in the intracellular (**C**) and extracellular compartments (**D**) determined by ^31^P MRS in the four melanoma xenografts are also shown. Only valuable time-dependent data were obtained for pHi and pHe in the DB-1 xenografts: pHi during trametinib treatment (i.e., y = −0.021x + 6.9, adjusted *R*^2^ = 0.240, and regression’s *p*-value = 0.01), and pHe in the DB-1 controls (i.e., y = -0.020x + 7.0, adjusted *R*^2^ = 0.235, and regression’s *p*-value = 0.02). In addition, the comparison of the slopes for pHe of DB-1 between untreated (m = -0.020) and trametinib-treated cells (m = 0.010) was statistically significant (*p* = 0.02). Notation in red denotes the adjusted *R*^2^ < 0.2 (low) or *p*-values that are non-significant (N.S.).

### 3.5. Assessment of Tumor Growth Following Trametinib Treatment

Figure 7 illustrates the changes in tumor volume over time in the different mouse xenografts. Untreated mice (control) showed the expected timely exponential and fast tumor size increase that occurs in cancer. Trametinib did not change this behavior in the WM3981 xenografts, but the remaining cell lines (WM983BR, WM983B, and DB-1) displayed a significantly reduced time-related tumor volume during therapy. However, while W983BR showed a time-related increase in volume (although not as fast as its untreated counterpart), WM983B and DB-1 showed a significant time-related decrease in tumor volume. As expected, the comparisons of the control vs. trametinib on each treatment day showed differences, but they were significant only on Day 5 of WM983B and Day 2 and Day 5 of the DB-1 xenografts.

**Figure 7 cells-13-01220-f007:**
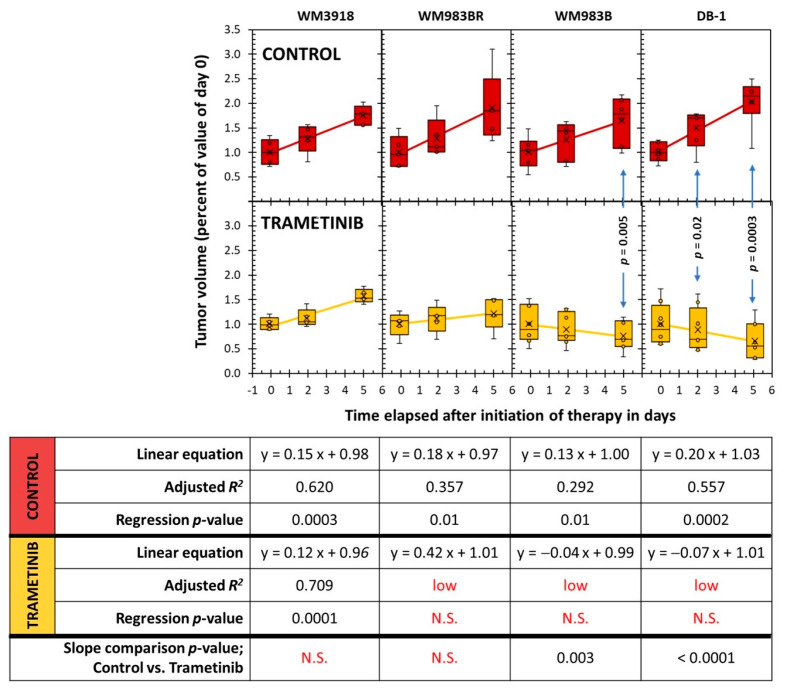
Comparison of tumor growth on Day 0, Day 2, and Day 5 between untreated controls (red) and trametinib-treated xenografts (yellow). The *p*-values in the graphs denote the significance of comparing the controls vs. trametinib-treated pairs at specific times. The table shows the parameters related to the statistical analysis of the data as time-dependent changes. Non-significant (N.S.).

## 4. Discussion

BRAF/MEK inhibition has become a standard-of-care option for *BRAF^V600^*-mutated melanoma. Dabrafenib, a BRAF inhibitor, and trametinib, a MEK inhibitor, have received approval from the U.S. Food and Drug Administration for treating BRAF-driven melanoma [21]. BRAF and MEK are crucial components of the MAPK signaling pathway, which regulates various essential cell functions, including cell growth and apoptosis [30,31,32]. We previously reported significant metabolic changes in preclinical melanoma models with mutated *BRAF* dependency sensitive to dabrafenib [10]. In the present work, we expanded these observations, demonstrating again the significant metabolomic phenotype variability in melanomas with high dependency on mutated *BRAF* [10,33,34]. However, different from our previous work inhibiting the altered BRAF protein directly using dabrafenib [10], in the present work, we use trametinib to inhibit MEK, which is a subsequent step after BRAF in the MAPK pathway (Figure 1). Thus, we reduced MAPK signaling by indirectly affecting the hyperactive BRAF protein, inhibiting MEK. Our findings strongly suggest a tight relationship between altered tumor metabolism and MAPK inhibition. Molecularly targeted agents inducing selective MEK inhibition play a crucial role in melanoma by inhibiting the abnormal MAPK signaling pathway and restoring apoptosis [35,36,37]. Hence, maximizing the benefits of inhibiting the MAPK pathway is crucial to achieving a high objective therapy response. Our noninvasive metabolic measurements (i.e., ^1^H and ^31^P MRS), which could be translatable to the clinical arena, may be valuable in determining the metabolic effects of BRAF and MEK inhibitors and, thus, could be effective clinical biomarkers of a response to these therapies. While this design allows for identifying early-treatment responses, it may not fully capture long-term metabolic adaptations or resistance mechanisms that could emerge over extended treatment periods.

We evaluated glucose uptake by isolated tumor cells, considering the correlation of this uptake with lactate production in cancer cells due to the Warburg effect [38]. Our results in Figure 2 demonstrate that trametinib reduced glucose uptake in all melanoma cell lines. However, the two sensitive cell lines showed reduced lactate production with trametinib, while the resistant ones significantly increased it. This decreased glucose uptake with increased lactate production may be due to the different energy sources the resistant cell lines utilize besides glucose (e.g., glutaminolysis, lipolysis, etc.). As an aside, these results could explain, at least in part, the difficulties of predicting the response via positron emission tomography (PET) [39,40,41]. PET is the modality for staging patients with solid malignancies, including melanoma [41,42,43]. For these purposes, the tumor uptake of the radiotracer ^18^F-2-fluoro-2-deoxy glucose (FDG) is followed by PET. FDG is an analog of glucose that cannot be metabolized further than the first step of glycolysis. As trametinib reduced the glucose uptake in all melanoma cell lines tested (Figure 2), an FDG-PET exam might not be able to demonstrate differences to predict the response. In contrast, ^1^H MRS showed lactate and alanine differences (Figure 5) associated with response and may be assessed noninvasively in melanoma patients.

Our results with the OCR and ECAR measurements (Figure 3 and Table 1) demonstrate that the wild-type cell line WM3918 under trametinib treatment showed no significant differences in OCR and the smallest, although statistically significant, decrease in ECAR of all cell lines compared to its control group. As expected, the OCR/ECAR did not change with trametinib therapy in WM3918. In comparison, the remaining cell lines showed significant but variable changes in OCR and ECAR. Trametinib proportionally increased both parameters in the resistant mutant WM983BR cell line, maintaining the OCR/ECAR constant at the lowest level of all cell lines. Both mutant cell lines, resistant (WM983BR) and sensitive (WM983B), had an increased OCAR but differed in ECAR. These changes demonstrate the upregulation of respiration in both cell lines, with an increase in glycolytic activity for WM983BR and a decrease in WM983B. Finally, the three parameters decreased in the remaining sensitive mutant cell line, DB-1. The trametinib-related changes in DB-1 demonstrate a decrease in respiratory and glycolytic activities, with a reduction in overall energy production.

Furthermore, the WM983B and DB-1 cell lines are BRAF-positive mutants, showing significant shifts in cell energy phenotype. These shifts reveal that MEK inhibition via trametinib impacts glycolytic activity in WM983B and DB-1. The trametinib-related decrease in ECAR in the sensitive mutant cell lines WM983B and DB-1 established them as less glycolytic (Figure 3). Figure 3 also shows that WM983B cells relied more on respiration with trametinib treatment and had the second-largest change in energy phenotype. DB-1, the most energetic melanoma line, was the most significantly affected and had the most prominent energy phenotype shift with treatment (Figure 3 and Table 1). These facts were corroborated by the decreased significant glucose uptake and lactate production shown in Figure 2 and the reduced tumor lactate values in vitro and in vivo depicted in Figure 4 and Figure 5, respectively. However, given that WM983B and WM983BR had a higher OCR with trametinib, they differed from DB-1, which had a lower OCR. Although we expected to see a reflection of these results in the bioenergetic status of the cell lines, the xenografts of these three cell lines had a trametinib-related increase in β-NTP/Pi (Figure 6). We theorize that despite the decreased cellular respiration found in DB-1 in the controlled in vitro OCR studies (Figure 3), its increased bioenergetic status in vivo could be due to the inhibition of the hyperactive MAPK pathway and the concomitant reduction in the Warburg effect. Under these in vivo conditions, instead of synthesizing lactate, glycolytic intermediates, mainly pyruvate, could be available for cellular respiration, although respiration seems to be deficient in DB-1 in the in vitro studies (Figure 3). The inhibition of the Warburg effect is supported by the substantial reduction in lactate production shown in Figure 2 and the reduction in tumor lactate content depicted in Figure 4 and Figure 5 for DB-1 and WM983B. However, it is also possible that the DB-1 inconsistencies may be due to the different experimental conditions of cultured tumor cells vs. in vivo xenografts.

The present results complement the ones we obtained on the effect of BRAF inhibition on the same melanoma models [10]. WM3918, the wild-type melanoma cell line, and WM983BR, the BRAF inhibitor-resistant mutant line, showed no significant shifts in OCR/ECAR ratio with trametinib treatment, indicating that the effect of trametinib was minimal. Other studies on MEK signaling inhibition in wild-type melanoma have shown similar findings [44]. However, WM3918 did show a significant drop in ECAR, which suggests that trametinib had some effect on glycolytic activity in the wild-type line. WM983BR showed small but significant increases in OCR and ECAR with trametinib treatment. Still, the OCR/ECAR ratio shift was not significant, probably due to high measurement deviations. Other studies on BRAF/MEK inhibition found that some BRAF mutants rapidly resisted BRAF and MEK inhibitors through genetic or epigenetic alterations [45]. In vitro resistance can also be seen in our results in the dabrafenib-resistant WM983BR. However, more work is needed to elucidate the discrepancies in the response of WM983BR to trametinib.

Our present results show that the cellular effects of a positive response to MEK inhibition were accompanied by a reduction in lactate production (Figure 2) and a time-dependent reduction in tumor lactate levels (Figure 4 and Figure 5). These results corroborate the report by Falck Miniotis et al. [14]. However, our methodology also allowed us to demonstrate changes in alanine, bioenergetics, and pH. As shown in Figure 5, the lactate changes matched alanine’s in WM983B and DB-1. However, reduced tumor lactate levels were found on Day 5, but alanine was not significantly reduced on that day in the WM983BR cell line. These data match the fact that WM983BR had a decreased but not significant tumor volume with trametinib treatment during our observation period, showing only a statistical trend (*p* = 0.06, Figure 7). This suboptimal response matches with a significant tumor lactate reduction but not with one for alanine, suggesting that WM983BR has a unique metabolic phenotype responding differently to trametinib compared to the two fully responding cell lines. Therefore, finding ways to induce an alanine reduction in WM983BR (e.g., using different therapies or combining other drugs with trametinib) may help improve its response. In comparison, WM983B and DB-1, the responding cell lines, showed the expected decrease in tumor growth while on trametinib therapy (Figure 7), correlating with significant lactate and alanine changes (Figure 5).

An essential objective of this work was to explore the clinical translatability of our standardized noninvasive tumor metabolic assessment. Experimental measurements using ^1^H MRS have revealed that the reduction in lactate observed after BRAF/MEK inhibition was more pronounced in cells containing the oncogenic *BRAF^V600E^* mutants than non-BRAF-driven cells [14]. These metabolic differences suggest that BRAF-independent cells do not experience a robust metabolic response to treatment, as lactate levels remained unchanged. However, a reduction in alanine was observed following MEK inhibition in the cell lines containing the oncogenic *BRAF^V600E^* mutation and in non-BRAF-driven cells (Figure 5). Alanine, a product of pyruvate amination and an essential component of the glutaminolysis pathway, plays a crucial role in cellular metabolism. Our findings in Figure 4 and Figure 5 demonstrate decreased lactate and alanine concentration with MEK inhibition, as observed by the in vitro and in vivo ^1^H MRS of preclinical melanoma models.

To demonstrate prediction by lactate and alanine, we adapted the clinically used RECIST (response evaluation criteria in solid tumors) [46,47] to determine the response using the mean group value of tumor volume on Day 5 of the trametinib treatment xenografts depicted in Figure 7. On Day 5, WM3918 had a 50% increase in tumor volume; thus, RECIST classifies it as a progressive disease (PD). In comparison, WM983BR had a tumor increase below 20%, and WM983B had a decrease of less than 30% in tumor volume, classifying both as stable diseases (SD). Finally, DB-1 showed a 40% decrease, so RECIST considers it a partial response (PR). Following this convention, the only xenograft with a positive response on Day 5 was DB-1 (PR). However, continued trametinib therapy could have brought some of these tumors to achieve a complete response (CR). The linear regressions for WM983B and DB-1 in Figure 7 predict that CR (i.e., the complete absence of a tumor) could have been achieved by Day 24 and Day 14, respectively. Notably, these two xenografts had significantly and sustainably reduced lactate and alanine since Day 2. In comparison, the time course of the tumor volume of WM983BR predicted that CR can never be reached. WM983BR had a significantly reduced tumor lactate value by Day 5, but the alanine value did not change (Figure 5). These analyses suggest that early and sustained reductions in tumor lactate and alanine values predict a subsequent CR in melanoma. They also suggest that a delayed reduction in lactate without changes in alanine (e.g., WM983BR) or no changes in both metabolites (e.g., WM3918) predict a negative response.

Using ^31^P MRS, we found that higher bioenergetic levels (i.e., β-NTP/Pi) correlate with an increased response to trametinib in responsive melanoma lines (Figure 6). However, only the DB-1-sensitive line showed a significant increase in extracellular pH (pHe). Tumors with higher glycolytic or energetic profiles, such as WM983B and DB-1, showed better responsiveness to trametinib than wild-type (WM3918) and BRAF-resistant (WM983BR) melanoma cell lines. The reason for this correlation between cellular energy state and glycolytic capacity remains unknown. However, this selective tumor de-energization could enhance the tumor’s response to therapeutic agents like trametinib.

Although we expected metabolic responses similar in trametinib compared to dabrafenib, as both affect the same regulatory pathway, we found subtle differences amongst the cell lines. For example, dabrafenib increased OCR significantly in WM983B, while the OCR change was insignificant with trametinib. Furthermore, when comparing the control with trametinib therapy, WM983BR has reduced (or inverted) time-related slopes for lactate and alanine in the xenograft studies. In comparison, both slopes were positive and not significantly different when using dabrafenib. The same WM983BR had a much steeper slope for the β–NTP/Pi increase in response to trametinib than dabrafenib [10].

Finally, the differences between the lactate levels before therapy could also be important as potential biomarkers of a response in patients with melanoma. The in vivo value of lactate in the untreated (control) DB-1 xenografts (the best responder to trametinib) was significantly larger in comparison to the rest of the xenografts (Figure 5). We hypothesize that finding a large tumor lactate value in melanoma patients before therapy could be predictive of a positive response with trametinib.

Our study demonstrates that trametinib treatment results in significant alterations in metabolic pathways in melanoma cells, evidenced by changes in alanine, glucose, lactate levels, and other metabolites. These changes suggest that trametinib may exert its effects through several potential mechanisms. Firstly, trametinib may reduce glycolytic flux, as indicated by decreased lactate production in treated cells. This lactate decrease could result from inhibiting MAPK/ERK pathway-regulated glycolytic enzymes. Secondly, the observed alterations in intracellular lactate and alanine levels suggest a shift in the metabolic balance between glycolysis and the TCA cycle. Trametinib may promote a metabolic shift towards oxidative phosphorylation by decreasing glycolysis, affecting cellular energy production and redox balance.

Additionally, the impact of trametinib on mitochondrial function, as reflected in our mitochondrial stress assays, points to a direct effect on oxidative phosphorylation. By inhibiting the MAPK/ERK pathway, trametinib may alter the expression and activity of mitochondrial enzymes, leading to reduced mitochondrial respiration and ATP production. Future research should aim to further elucidate these mechanisms through targeted metabolomic studies and genetic analyses. For instance, assessing key metabolic enzymes’ and transporters’ expression and activity in response to trametinib treatment could provide deeper insights into the drug’s impact on cellular metabolism. Additionally, the genetic manipulation of specific metabolic pathways could help identify critical nodes mediating trametinib’s metabolic effects. Overall, understanding the mechanistic basis of trametinib’s metabolic effects will enhance the therapeutic targeting of melanoma and potentially identify novel biomarkers for treatment response. These biomarkers could help improve precision medicine for melanoma patients by providing clinicians with advanced tools to predict treatment responses better, monitor disease progression, and establish personalized therapeutic strategies.

One of the critical limitations of this study is the inherent tumor heterogeneity observed in melanoma. Tumor heterogeneity refers to diverse cell populations within a single tumor and between different tumors in the same patient. This diversity can arise from genetic, epigenetic, and environmental differences, leading to tumor behavior, treatment response, and disease progression variations. The reliance on human melanoma xenograft models in athymic nude mice for melanoma research presents certain limitations. While these models are valuable for studying tumor growth and treatment effects in a controlled environment, they do not fully mimic the complexity of human melanoma physiology. Differences in immune system function, tumor microenvironments, and genetic backgrounds between mice and humans can result in distinct responses to therapies.

Consequently, the findings obtained from mouse xenograft studies may not be directly applicable or fully predictive of how treatments will perform in human patients. This discrepancy can limit the translatability of preclinical results to clinical settings, where the goal is to improve patient outcomes and develop effective therapies for human melanoma. Another limitation of this study is the resolution of MRS. While MRS is a powerful, noninvasive tool for studying metabolic changes in vivo, it has inherent limitations in spatial and spectral resolution and sensitivity, generating technical challenges that can affect the accuracy and reliability of the data obtained.

## 5. Conclusions

Upon reducing hyperactive MAPK signaling using trametinib, we found diverse metabolomic changes in phenotypically diverse models of melanoma. Notably, these metabolic changes are correlated with the response to the inhibitor (i.e., reduction in tumor growth) and thus have the potential to be used as biomarkers of prediction or follow-up in melanoma patients. Our results suggest that the noninvasive ^1^H/^31^P MRS assessment of lactate, alanine, bioenergetics levels (β-NTP/Pi), and pH (pHi and pHe), especially when complemented with multiparametric MR imaging of the tumor microenvironment (e.g., vascularity and cellularity) and FDG-PET, will offer valuable information for the personalized medicine approach to treating patients with melanoma. Our current study provides proof of principle for investigating lactate and alanine metabolism, bioenergetics levels (β-NTP/Pi), and pH using MRS in BRAF/MEK inhibitor-treated patients with metastatic melanoma and possibly other solid tumors.

This study evaluated the metabolic effects of trametinib, an MEK inhibitor, in preclinical human melanoma models. Using ^1^H and ^31^P MRS and complementary biochemical assays, we observed significant changes in tumor metabolomics, including modifications in lactate and alanine levels and bioenergetics. These changes were more pronounced in cells with the *BRAF^V600E^* mutation, indicating a more significant response to trametinib. Additionally, trametinib treatment decreased tumor volume in both wild-type and mutant melanoma xenografts, with the most considerable effects observed in sensitive cell lines. Biochemical measurements of glycolysis and bioenergetics could provide valuable predictive estimates of melanoma patients’ responses to treatment, particularly in assessing the energetic demands of therapy. This evidence may also apply to monitoring and predicting responses in other cancers and treatment modalities, including BRAF, MEK inhibitors, and immunotherapy. The study highlights the potential of metabolic imaging techniques like MRS in assessing treatment response and guiding therapy in melanoma patients.

## Data Availability

The dataset is available on request from the authors.

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
