# Peer review of "1H and 31P Magnetic Resonance Spectroscopic Metabolomic Imaging: Assessing Mitogen-Activated Protein Kinase Inhibition in Melanoma"

_cells, 2024, doi:10.3390/cells13141220_

Round 1

Reviewer 1 Report

Comments and Suggestions for Authors

This study shows that 1H MRS detects metabolic changes, like reduced lactate and alanine levels, correlating with treatment response in preclinical melanoma models. It supports using noninvasive MRS to predict early melanoma response to MEK inhibition. The MAPK pathway, with BRAF mutations, drives 40-60% of melanomas. Current BRAF and MEK inhibitors are not always effective.

1. The authors utilized 0.1% DMSO as a control, but it would have been beneficial to include a negative control without DMSO to isolate the effects of DMSO. Ensuring that the 0.1% DMSO used does not independently affect the results is crucial, despite its minimal concentration.

2. The reliance on xenograft models in mice may not fully replicate human melanoma physiology and treatment responses. This could limit the translatability of findings to clinical settings where patient outcomes are ultimately measured. It would be beneficial for the author to address this issue in the text.

3. Typically, these studies can last from several weeks to several months. This study's short duration of trametinib treatment (five days) in animal models may not capture long-term metabolic adaptations or resistance mechanisms that could develop over prolonged treatment periods in human patients. Of course, it depends on the tumor size and volume (from animal study).

4. While the detailed experimental protocols are commendable for scientific rigor, the complexity of methodologies like high-resolution 1H-MRS and mitochondrial stress assays may pose challenges in reproducibility and accessibility for other researchers without specialized equipment or expertise.

5. The choice of linear regression for analyzing time-related changes, despite expectations of exponential behavior in biological data such as tumor growth, may oversimplify the interpretation of metabolic responses to treatment. This could potentially obscure nuanced metabolic dynamics.

6. While the study focuses on glucose, lactate, and select metabolites, broader coverage of metabolic pathways or additional biomarkers could provide a more comprehensive understanding of trametinib's effects on cellular metabolism and treatment response.

7. Some inconsistencies in metabolic responses between in vitro and in vivo models (e.g., trametinib's effect on lactate levels in WM3918 cells) are observed, suggesting the need for further validation and understanding of these discrepancies.

8. The interpretation of results in some figures (e.g., Figure 5) could be clearer, particularly regarding the statistical significance of observed differences.

9. Limited discussion on the mechanistic insights into how trametinib alters metabolic pathways in melanoma cells, which could enhance the study's impact and relevance.

10. What is the meaning of blank in the comparison table?

Author Response

Please find attached the reply to the review report (Reviewer 1).

Reviewer 2 Report

Comments and Suggestions for Authors

In this manuscript, Gupta et al. studied the metabolic changes during the treatment of human melanoma model by using the magnetic resonance spectroscopy. They constructed both the in-vitro cell model and the in-vivo mice model with four human melanoma cell lines and investigated the metabolic effects of trametinib. Nevertheless, this manuscript was too similar to their previously published paper (Cancers 2024, 16, 365.), which greatly reduced its novelty. Besides, the content is not abundant enough, and the conclusion is not significant, which did not give a definite discovery of new mechanism or practical case. Thus, the reviewer can’t support the publication of this manuscript at the current stage. The major issues are listed as following:

1. In the title, the “metabolic imaging biomarkers” is confusing. The reviewer didn’t see any imaging results. Although the author used the magnetic resonance spectroscopy, but they did not provide any images of the cells or mice.

2. Comparing to their previously published paper (Cancers 2024, 16, 365.), the authors totally used the same four cell lines and same analytical methods. Even the title and figures are extremely similar. The reviewer thought a significant adjustment of the paper structure was necessary.

3. The reliability the in-vivo magnetic resonance spectroscopy results need to be demonstrated by other traditional methods. Besides, the advantages of using magnetic resonance spectroscopy should be clarified more clearly.

4. Some abbreviations should be defined at their first appearance, such as BRAF, MEK, RAS. Some characters in Figure 1B and F are too small to distinguish.

Author Response

Please find attached the reply to the review report (Reviewer 2).

Reviewer 3 Report

Comments and Suggestions for Authors

The authors follow up on a previous study which used magnetic resonance spectroscopy to observe changes in metabolic signatures of melanoma cells treated with the BRAF inhibitor dabrafenib. In the curreny study, four melanoma cell lines (two sensitive to BRAF inhibitors) are examined using MRS before and after treatment with the MEK inhibitor trametinib. This serves as a kind of pilot or proof of principle for using noninvasive MRS to detect biomarkers of treatment response.

Interestingly, melanoma cell lines had decreases in lactate production following trametinib treatment only if they were sensitive to BRAF inhibitors. OCR vs ECR treatment-induced changes for these cell lines were less distinct. Lactate and alanine levels (by MRS) in cell lines seem equivocal for alanine, with treatment inducing significant decreases regardless of BRAF status, but with lactate decreasing only in the sensitive lines.

As xenografts in mice, treatment reduced both lactate and alanine over time in the sensitive xenografts. Sensitive xenografts had subtly higher energetics (B-NTP) over time in the untreated conditions. Intracellular and extracellular pH changes were only significant in one untreated sensitive xenograft.

On the whole, the authors do demonstrate that MRS can detect differences in response to treatment, and therefor may serve as a prognostic tool with lactate and alanine as potential biomarkers. The novelty is lessened by the previous work showing essentially this, but for dabrafenib instead of trametinib, but this is a reasonable continuation of that work.  Also lessening the impact is the small number of cell lines under examination.  Follow up work in human subjects would be particularly useful to demonstrate whether or not this method is useful as a therapeutic guide, and this translation (as the authors mention) should be straightforward as this method is noninvasive.

In all, the study, as a proof of principle, is interesting and well written. Methods are appropriately detailed and claims are generally appropriately hedged. The discussion could benefit from a paragraph about the limitations of the study and future directions.

Author Response

Please find attached the reply to the review report (Reviewer 3).

Round 2

Reviewer 1 Report

Comments and Suggestions for Authors

The author has effectively addressed the comments provided. They have made sure to incorporate the suggested revisions comprehensively, ensuring that the points raised were well integrated into the text. This thorough response demonstrates their attention to detail and commitment to improving the manuscript based on constructive feedback.

Reviewer 2 Report

Comments and Suggestions for Authors

The authors have made significant improvements in the revised manuscript, now the reviewer support the publication of this manuscript.